# Impact of Peptidoglycan Recycling Blockade and Expression of Horizontally Acquired β-Lactamases on *Pseudomonas aeruginosa* Virulence

Isabel M. Barceló,[a,b] Gabriel Torrens,[a,b] María Escobar-Salom,[a,b] Elena Jordana-Lluch,[a] María Magdalena Capó-Bauzá,[a] Carlos Ramón-Pallín,[a] Daniel García-Cuaresma,[a] Pablo A. Fraile-Ribot,[a,b] Xavier Mulet,[a,b] (iD) Antonio Oliver,[a,b] (iD) Carlos Juan[a,b]

aMicrobiology Department and Research Unit, University Hospital Son Espases, Health Research Institute of the Balearic Islands (IdISBa), Palma, Spain
bCIBER de Enfermedades Infecciosas, Madrid, Spain

Isabel M. Barceló and Gabriel Torrens contributed equally to this article. Author order was determined alphabetically.

**ABSTRACT** In the current scenario of antibiotic resistance magnification, new weapons against top nosocomial pathogens like *Pseudomonas aeruginosa* are urgently needed. The interplay between β-lactam resistance and virulence is considered a promising source of targets to be attacked by antivirulence therapies, and in this regard, we previously showed that a peptidoglycan recycling blockade dramatically attenuated the pathogenic power of *P. aeruginosa* strains hyperproducing the chromosomal β-lactamase AmpC. Here, we sought to ascertain whether this observation could be applicable to other β-lactamases. To do so, *P. aeruginosa* wild-type or peptidoglycan recycling-defective strains (Δ*ampG* and Δ*nagZ*) harboring different cloned β-lactamases (transferable GES, VIM, and OXA types) were used to assess their virulence in *Galleria mellonella* larvae by determining 50% lethal doses ($LD_{50}$s). A mild yet significant $LD_{50}$ increase was observed after peptidoglycan recycling disruption *per se*, whereas the expression of class A and B enzymes did not impact virulence. While the production of the narrow-spectrum class D OXA-2 entailed a slight attenuation, its extended-spectrum derivatives OXA-226 (W159R [bearing a change of W to R at position 159]), OXA-161 (N148D), and principally, OXA-539 (D149 duplication) were associated with outstanding virulence impairments, especially in recycling-defective backgrounds (with some $LD_{50}$s being >1,000-fold that of the wild type). Although their exact molecular bases remain to be deciphered, these results suggest that mutations affecting the catalytic center and, therefore, the hydrolytic spectrum of OXA-2-derived enzymes also drastically impact the pathogenic power of *P. aeruginosa*. This work provides new and relevant knowledge to the complex topic of the interplay between the production of β-lactamases and virulence that could be useful to build future therapeutic strategies against *P. aeruginosa*.

**IMPORTANCE** *Pseudomonas aeruginosa* is one of the leading nosocomial pathogens whose growing resistance makes the development of therapeutic options extremely urgent. The resistance-virulence interplay has classically aroused researchers' interest as a source of therapeutic targets. In this regard, we describe a wide array of virulence attenuations associated with different transferable β-lactamases, among which the production of OXA-2-derived extended-spectrum β-lactamases stood out as a dramatic handicap for pathogenesis, likely as a side effect of mutations causing the expansion of their hydrolytic spectrums. Moreover, our results confirm the validity of disturbing peptidoglycan recycling as a weapon to attenuate *P. aeruginosa* virulence in class C and D β-lactamase production backgrounds. In the current scenario of dissemination of horizontally acquired β-lactamases, this work brings out new data on the complex interplay between the production of specific enzymes and virulence attenuation that, if complemented with the characterization of the underlying mechanisms, will likely be exploitable to develop future virulence-targeting antipseudomonal strategies.

Address correspondence to Antonio Oliver, antonio.oliver@ssib.es, or Carlos Juan, carlos.juan@ssib.es.

The authors declare no conflict of interest.

**KEYWORDS** *Pseudomonas aeruginosa*, peptidoglycan recycling, AmpG, NagZ, AmpC cephalosporinase, horizontally acquired $\beta$-lactamases, $bla_{VIM-1}$, $bla_{GES-1}$, $bla_{OXA-2}$-like $\beta$-lactamases, extended-spectrum $\beta$-lactamases, virulence, *Galleria mellonella*

*P*seudomonas aeruginosa is one of the foremost nosocomial pathogens, standing out as the etiologic agent of acute opportunistic infections (urinary tract, ventilator-associated pneumonia, burns, bacteremia, etc.) and chronic infections in patients with underlying respiratory diseases like cystic fibrosis (1, 2). The most defining feature of this species is its capacity for antibiotic resistance development through multiple mechanisms (such as the peptidoglycan metabolism-linked AmpC intrinsic cephalosporinase or the transferable $\beta$-lactamases) that progressively weaken our therapeutic arsenal (3, 4). *P. aeruginosa* is one of the species considered a "critical priority" by the WHO, i.e., the group of pathogens for which the development of new treatments is most urgent (5). In this regard, besides some relatively new formulations and combinations with $\beta$-lactamase inhibitors (6), other imaginative options, such as the so-called antivirulence therapies, are gaining importance as potential antipseudomonal weapons for the future. Their goal is to decrease bacterial pathogenic power, thereby reducing the clinical consequences for the patient and gaining chances of infection clearance by the immune system (7–9). In this regard, the interplay between resistance and virulence has classically aroused interest as a source of exploitable weak points to base these kinds of therapies on (10, 11), and we have made some findings in the field, such as the dramatic loss of virulence in *P. aeruginosa* after the combination of AmpC hyperproduction plus a peptidoglycan recycling blockade (12). In the light of these results, we hypothesize that AmpC, when overproduced, may drive side effects on bacterial biology—only visible if peptidoglycan recycling is disturbed—that impair the pathogenic power of *P. aeruginosa*, effects that could potentially be caused by other $\beta$-lactamases too. To verify this hypothesis, in the present study, we analyze whether this virulence attenuation is reproduced when *P. aeruginosa* expresses different horizontally transferable $\beta$-lactamases instead of solely AmpC. This analysis is not only suitable to understand whether potentially shared features among different $\beta$-lactamases can enable the same dynamics of virulence loss but also timely given the clinical-epidemiological threat that the dissemination of transferable $\beta$-lactamases poses worldwide (13). In this respect, although the array of horizontal $\beta$-lactamases described to date in *P. aeruginosa* is almost endless (14), some of the representatives most usually found in clinical isolates are certain narrow- and extended-spectrum $\beta$-lactamases (ESBLs) belonging to class A (such as GES enzymes) and class D (such as OXA variants), as well as certain class B carbapenemases (such as those from the IMP and VIM families, for instance) (15, 16). More specifically, some OXA enzymes are of additional interest since their plasticity for adaptation to different substrates has repeatedly and recently been demonstrated, including acquiring mutations conferring resistance to even new cephalosporin–$\beta$-lactamase inhibitor combinations, such as ceftolozane-tazobactam or ceftazidime-avibactam. In this regard, we have previously described in *P. aeruginosa* three OXA-2-derived ESBLs—OXA-226 (W159R [bearing a change from W to R at position 159]), OXA-161 (N148D), and OXA-539 (D149 duplication)—that pose an excellent model for understanding how specific amino acid changes in their catalytic centers can determine not only the enzymes' hydrolytic profile but also potentially different impacts on virulence, a topic we wanted to go further into with this study (17–19).

In summary, here, we characterize a collection of *P. aeruginosa* strains (wild type versus AmpG- or NagZ-defective mutants as models for peptidoglycan recycling disturbance) (20, 21) harboring a wide array of cloned transferable $\beta$-lactamases (class A, B, or D enzymes) in order to determine their killing capacities against *Galleria mellonella* larvae as a general measure of virulence. We also analyze different indicators linked to pathogenesis (growth rate, motility, cell culture infection-related parameters, etc.) to provide an idea of the potential pathways leading to virulence loss. In brief, our results reveal important differences between degrees of attenuation depending on the $\beta$-lactamase and

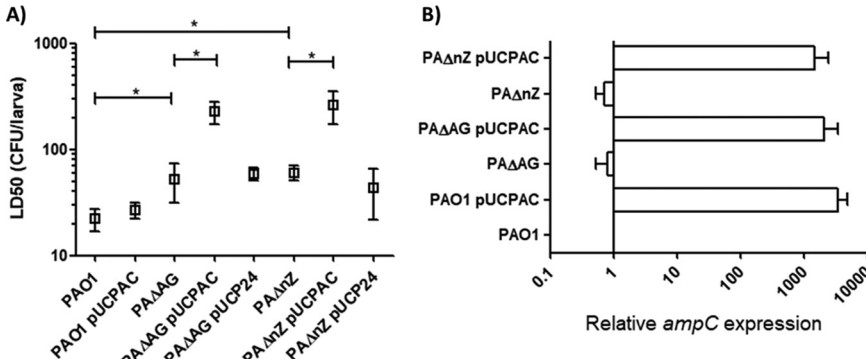

**FIG 1** (A) *Galleria mellonella* killing assays after infection with strain PAO1 or mutants with impaired peptidoglycan recycling, expressing or not expressing the AmpC cephalosporinase cloned in the plasmid pUCP24. Experiments and calculations of $LD_{50}s$ were performed as explained in Materials and Methods. The mean values of calculated $LD_{50}s$ and the standard deviations (SDs) obtained from at least three independent experiments are represented by boxes and error bars, respectively. All data are displayed on a log scale. *, $P < 0.05$ in Student's *t* test for comparisons between calculated $LD_{50}s$. Mutants harboring the empty pUCP24 vector were also included as controls, but the obvious statistical significance obtained when comparing their $LD_{50}s$ with those of the corresponding strains harboring pUCPAC is not shown in order to lighten up the figure. (B) Relative changes in *ampC* mRNA (considering PAO1 expression as 1) in the peptidoglycan recycling-impaired strains or in those expressing the *ampC* $\beta$-lactamase cloned in the pUCP24 multicopy vector. Horizontal open bars represent the mean values from experimental replicates, whereas the error bars correspond to the SDs. All data are displayed on a log scale.

suggest that the bases underlying these virulence impairments are intimately related to the mutations in the catalytic center responsible for the expansion of the hydrolytic spectrum of OXA-2. Thus, these facts draw an intricate panorama of mechanisms through which the expression of specific $\beta$-lactamases and/or peptidoglycan recycling disturbance dampen the pathogenic power of *P. aeruginosa*. Our work provides new data contributing to increased knowledge in the field, worthy of being delved into further in order to design future antivirulence weapons against *P. aeruginosa*.

## RESULTS

**Peptidoglycan recycling impairment attenuates *P. aeruginosa*'s killing capacity against *Galleria mellonella*.** As can be seen by the results in Fig. 1A and according to what we described in the past (12), the combination of a peptidoglycan recycling blockade (achieved through *ampG* inactivation) plus AmpC hyperproduction (through its expression from the multicopy plasmid pUCPAC) causes a dramatic loss of virulence in *P. aeruginosa*: the $LD_{50}$ of strain PA$\Delta$AG(pUCPAC) on *G. mellonella* larvae was ca. 225 CFU, whereas that of PAO1 was ca. 25 CFU. As a novel finding, we also show that in a background of peptidoglycan recycling impairment through *nagZ* disruption, attenuation of virulence appears to a similar extent when AmpC is concomitantly hyperproduced (the $LD_{50}$ increased ca. 10-fold compared to that of PAO1). Conversely, the hyperproduction of AmpC *per se* (meaning thousands-fold compared to the expression level in PAO1) (Fig. 1B) (12) was not enough to cause a statistically significant impact on the killing capacity of *P. aeruginosa*. Meanwhile, our results indicate that peptidoglycan recycling impairment *per se* caused slight yet statistically significant increases in $LD_{50}s$, with the strain PA$\Delta$nZ and strain PA$\Delta$AG values being ca. 2-fold that of PAO1. Finally, as can be seen by the results in Fig. 1B, the levels of expression of *ampC* in the different strains harboring pUCPAC were similar, ruling out the possibility of the various levels of attenuation shown in Fig. 1A being caused by differential levels of $\beta$-lactamase production and, therefore, different energy costs in each strain.

**Production of the class D OXA-2 $\beta$-lactamase, but not that of class A (GES-1) and B (VIM-1) enzymes, attenuates *P. aeruginosa* virulence.** As previously shown (12) and as corroborated by our results here, AmpC hyperexpression happening in a peptidoglycan recycling-impaired background entails a drastic virulence loss for *P. aeruginosa*. To go further into the bases and implications of this outcome, we sought to ascertain whether this

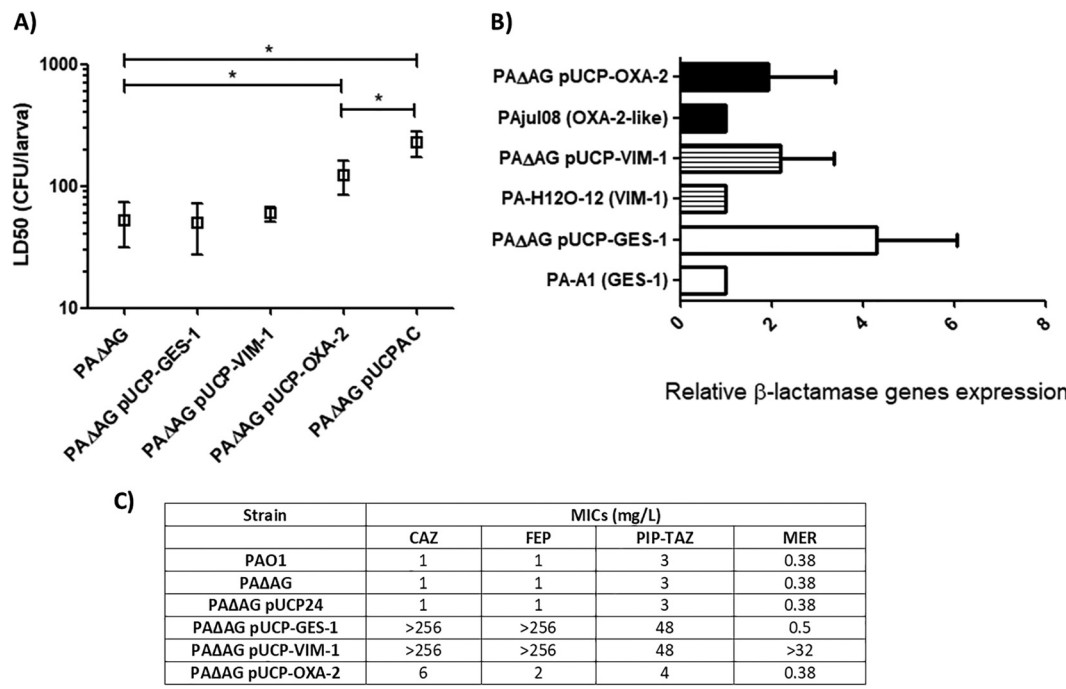

**FIG 2** (A) *Galleria mellonella* killing assays with the AmpG knockout mutant expressing different transferable β-lactamases cloned in the plasmid pUCP24. Mean values ± SDs obtained from at least three independent experiments are represented by boxes and error bars. All data are displayed on a log scale. (B) Relative increase in the mRNA of each β-lactamase gene cloned in the pUCP24 vector and transformed into the AmpG-defective mutant (considering the respective control strains' expression as 1). Horizontal bars represent the mean values from experimental replicates, whereas the error bars correspond to the SDs (linear scale). Patterns were assigned to facilitate the recognition of each measured mRNA, as follows: black, $bla_{OXA-2}$-like; lined pattern, $bla_{VIM-1}$; white, $bla_{GES-1}$. (C) MICs of representative β-lactams (obtained through commercial test strips) against the PAΔAG mutants expressing different β-lactamases cloned in the pUCP24 multicopy vector. Appropriate control values are also included for comparison. CAZ, ceftazidime; FEP, cefepime; PIP-TAZ, piperacillin-tazobactam; MER, meropenem.

attenuation could be reproduced when the bacterium expressed other β-lactamases or if, conversely, it was specifically associated with AmpC. For this purpose, we analyzed the *G. mellonella* killing capacity of the AmpG knockout (KO) mutant (as a model of a peptidoglycan recycling-impaired strain) expressing three representatives of horizontally transmitted β-lactamase types very commonly found in clinical isolates, i.e., the class A ESBL GES-1, the class B carbapenemase VIM-1, and the narrow-spectrum class D OXA-2 enzyme (13, 14, 22, 23). As shown by the results in Fig. 2A, the expression of the first two enzymes did not have any significant impact beyond the slight $LD_{50}$ increase caused by AmpG inactivation. Conversely, when OXA-2 was expressed in the AmpG-defective mutant, a significant increase in the $LD_{50}$ was obtained (ca. 2-fold compared to that of strain PAΔAG devoid of the plasmid), not far from the value of strain PAΔAG(pUCPAC). Therefore, these results suggest that the attenuation of virulence associated with *ampC* hyperexpression in a recycling-defective background is linked to certain effects caused by this class C cephalosporinase—potentially also attributable to OXA-2—that are not in common for the class A and B β-lactamases studied.

To verify that the strains harboring pUCP-GES-1 and pUCP-VIM-1 effectively produced the respective enzyme, we quantified the expression of their codifying genes by comparing it with that of two β-lactam-resistant clinical strains for which the expression of each enzyme was expected to be relevant (24, 25). The same was carried out with OXA-2, comparing its expression from the plasmid with that of the previously described clinical strain PAjul08 (17). As can be seen by the results in Fig. 2B, the levels of expression of the cloned $bla_{VIM-1}$ and $bla_{GES-1}$ genes were even slightly higher than those of the respective clinical strains, suggesting that the absence of attenuation linked to these β-lactamases was not due to a lack of their production. The representative β-lactams' MICs

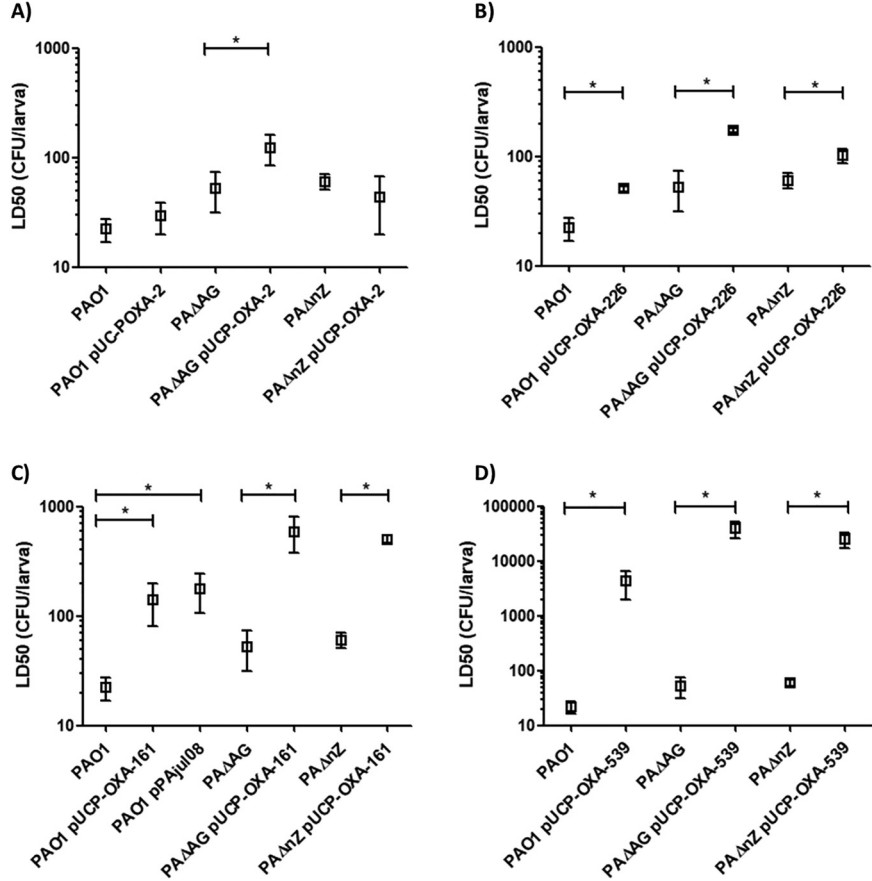

**FIG 3** *Galleria mellonella* killing assays with the PAO1 wild-type strain and AmpG- and NagZ-defective mutants expressing OXA-2 β-lactamase or its ESBL derivatives cloned in the pUCP24 plasmid. Mean values ± SDs obtained from at least three independent experiments are represented by boxes and error bars. All data are displayed on a log scale. *, $P < 0.05$ in Student's *t* test for comparisons between calculated $LD_{50}$s. Statistical significance was only analyzed for comparisons between each control strain (PAO1, PAΔAG, or PAΔnZ) and the same strain containing the plasmid with the corresponding cloned enzyme. (A) Killing assays with strains containing pUCP-OXA-2 and their respective controls. (B) Killing assays with strains containing pUCP-OXA-226 and their respective controls. (C) Killing assays with strains containing pUCP-OXA-161 and their respective controls. In this case, PAO1 harboring the natural plasmid from which $bla_{OXA-161}$ was first described (17) was also included. (D) Killing assays with strains containing pUCP-OXA-539 and their respective controls.

for the strains harboring each β-lactamase (Fig. 2C) also corroborate the effective production of each enzyme. In summary, it can be deduced that the above-mentioned virulence attenuations are not exclusively due to the energy burden of producing whichever β-lactamase was harbored or to common effects produced by all these enzymes on bacterial biology but, rather, to specific features of AmpC and perhaps other specific β-lactamases, such as OXA-2.

Given the hints provided by the increased $LD_{50}$ of strain PAΔAG(pUCP-OXA-2), we sought to extend our analysis to better understand the impact on the virulence of this specific enzyme and its derivatives. For this purpose, we first analyzed whether the decrease in virulence associated with OXA-2 could be reproduced in other backgrounds (strain PAO1 and NagZ-defective strains). Our results displayed in Fig. 3A indicate that when this enzyme was produced in the wild-type and NagZ-defective strains, no significant changes in the $LD_{50}$s occurred compared to the those of the respective background strain devoid of pUCP-OXA-2. This contrasts with the cited increase of the $LD_{50}$ of ca. 2-fold when the enzyme was produced in strain PAΔAG, which indicates that although OXA-2 expression may entail a certain biological cost, it is only seen under conditions in which peptidoglycan recycling is expected to be virtually blocked

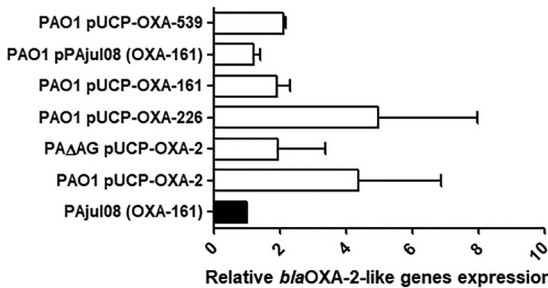

**FIG 4** Relative increases in the mRNA of $bla_{OXA-2}$ or derivative genes cloned in the pUCP24 vector and transformed into strain PAO1 (or strain PAΔAG), considering the expression of $bla_{OXA-161}$ from the clinical strain PAjul08 (17) (black bar) as 1. Horizontal bars represent mean values from experimental replicates, whereas the error bars correspond to SDs (linear scale).

(i.e., the disruption of AmpG, the main gate for the entrance of peptidoglycan fragments into the cytosol). Conversely, this handicap was not observable in the wild-type strain or in the NagZ-defective mutant (for which recycling impairment is thought to be not as disturbed as that of strain PAΔAG [26]), suggesting a negligible burden associated with the expression of this narrow-spectrum enzyme under regular conditions.

**Mutations leading to extended-spectrum OXA-2-type β-lactamases drastically impair *P. aeruginosa*'s virulence.** Given their growing relevance, we further extended our study to previously described OXA-2 derivatives containing key mutations in the catalytic center leading to ESBL phenotypes, such as OXA-226 (W159R), OXA-161 (N148D), and OXA-539 (D149 duplication) (17–19). As shown by the results in Fig. 3B to D, in contrast to OXA-2 production, the expression of OXA-226, OXA-161, and especially OXA-539 had a major impact on virulence, involving dramatic increases in the $LD_{50}$s even in the PAO1 strain, although more exaggerated in backgrounds with disturbed peptidoglycan recycling. In the case of pUCP-OXA-226, the $LD_{50}$s were ca. 50 CFU for PAO1, 173 CFU for PAΔAG, and 102 CFU for PAΔnZ (Fig. 3B). In the case of pUCP-OXA-161, the $LD_{50}$s were 140 CFU for PAO1, 590 CFU for PAΔAG, and 501 CFU for PAΔnZ (Fig. 3C). Therefore, the expression of OXA-161 entailed a higher cost even in the wild-type background, which was confirmed in the PAO1 strain carrying the natural plasmid harboring $bla_{OXA-161}$ (17). Strikingly, the $LD_{50}$s were much higher for the strains producing OXA-539, which had extraordinarily elevated $LD_{50}$s: ca. 4,300, 38,500, and 25,000 CFU for strains PAO1, PAΔAG, and PAΔnZ, respectively (Fig. 3D).

To rule out the possibility of these differences in levels of attenuation being due to discrepancies in the levels of β-lactamase production, we quantified the respective mRNAs by comparing them with the expression in a clinical strain expressing $bla_{OXA-161}$ (PAjul08). Since $bla_{OXA-2}$, $bla_{OXA-226}$, $bla_{OXA-161}$, and $bla_{OXA-539}$ only differ in punctual nucleotides (one to three) (17–19), the cited clinical strain was set as the unique reference and the same primers were used for real time RT-PCR (RT-PCR) after checking the absence of changes at the hybridization sites. As can be seen from the results in Fig. 4, the levels of expression of the four OXA genes were very similar. As expected, the background in which the plasmid was produced did not have any significant impact on β-lactamase expression either. These facts suggest that the different levels of virulence loss we show here do not respond to discrepancies in the amount of enzyme produced but, rather, to specific features and potential side effects of each OXA-2-like β-lactamase on bacterial biology.

**Investigating the bases for the virulence attenuation associated with the production of transferable OXA-2-type β-lactamases.** Here, we characterized our strains regarding some generally used fitness- and virulence-related parameters (12), to ascertain whether these could provide some information about the mechanisms underlying the virulence attenuation profiles described above. In the experiments described in this section, besides the wild-type PAO1, only the AmpG-defective mutant expressing the different cloned β-lactamases was used, since it was the background in which the OXA-linked virulence attenuations were more exaggerated and was therefore

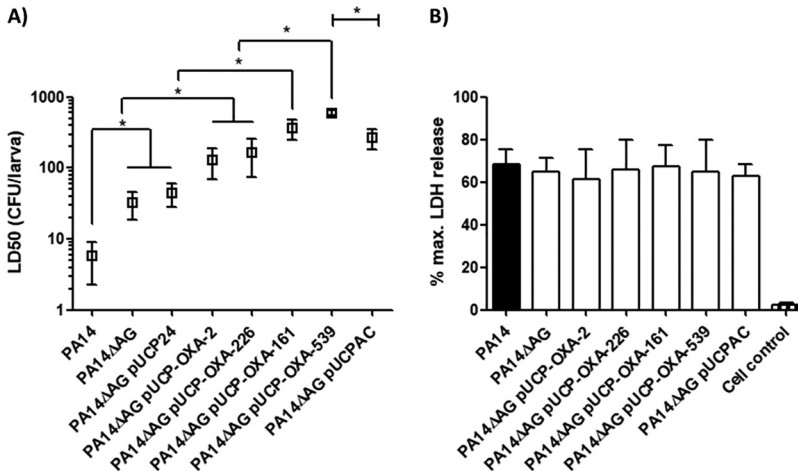

**FIG 5** (A) *Galleria mellonella* killing assays with the PA14 wild-type or AmpG-defective strain expressing the OXA-2 β-lactamase or its ESBL derivatives cloned in the pUCP24 plasmid. Mean values ± SDs obtained from at least three independent experiments are represented by boxes and error bars. All data are displayed on a log scale. *, $P < 0.05$ in Tukey's *post hoc* test for multiple comparisons between calculated LD$_{50}$s. Strains are grouped with horizontal uncapped lines when no statistical difference exists between them, whereas the symbols for obvious statistical significances have been excluded in order to lighten up the figure. The PA14ΔAG mutant harboring the empty pUCP24 vector or pUCPAC was also included as a control. (B) Cytotoxicity (LDH release) results after infection (MOI of 100, 3 h) on A549 cells. Results (mean value ± SDs) are expressed as the percentages with regard to the maximum LDH that can be released, i.e., from a well of completely lysed confluent A549 cells. The percentage of LDH released by uninfected cells is also shown as a control.

chosen as the model. To start with, we determined the exponential duplication times of the different strains indicated, and as can be seen by the results in Fig. S1, although significant increases in the values were associated with the presence of cloned β-lactamases, there was no correlation with those linked to greater virulence attenuations. In fact, peptidoglycan recycling disturbance *per* se (duplication time of 33.8 min for strain PAO1 versus 36.9 min for strain PAΔAG) or the sole presence of pUCP24 [duplication time of 40.9 min for PAΔAG (pUCP24)] increased the duplication times in comparison to those of the respective controls. Meanwhile, the presence of any of the cloned β-lactamases only caused, at most, a slight but not statistically significant increase in duplication time compared to that of the background strain with the empty vector. Therefore, our results suggested that growth rate was not the basis for the differential levels of attenuation linked to the enzymes studied here.

Regarding the cytotoxicity parameter, since PAO1 is barely cytotoxic in cell culture, the highly virulent PA14 strain has been repeatedly used in the past to approach its characterization (12). Thus, we used PA14 and its derived AmpG mutant harboring the different OXA-2-like enzymes, first to verify attenuations of virulence on *G. mellonella* and afterwards to check whether cytotoxicity against A549 cells was affected as well (Fig. 5). Despite starting from different initial LD$_{50}$s, since PA14 is more virulent (LD$_{50}$ of ca. 5 CFU) (Fig. 5A) (27), the dynamics of virulence loss associated with the peptidoglycan recycling disruption and with the production of OXA-2-like β-lactamases were similar to those reported for PAO1, revealing very significant increases in the LD$_{50}$s principally for OXA-539 [reaching ca. 600 CFU, a value even higher than that of PA14ΔAG(pUCPAC)]. On the other hand, the results concerning cytotoxicity profiles indicate that the decreases in virulence associated with the production of OXA-2-like or AmpC enzymes in an AmpG-defective background are not mediated by a reduction in this feature (all the strains showed values between 60 and 70% of maximum lactate dehydrogenase [LDH] release) (Fig. 5B), and consequently, other molecular explanations must exist. Apparently, these potential bases are not related to either a disturbed capacity for cellular adhesion/invasion (Fig. S2A) or to alterations in the inflammatory response triggered in infected cells (Fig. S2B), since the results for all the strains studied were similar, with some strains with punctual differences never reaching the significance threshold (P value of >0.05 determined by analysis of variance [ANOVA]).

In dealing with motility (Fig. S3), no differential impacts on this feature were found to be linked to the expression of the cloned enzymes, with the differences that were found never being statistically significant (*P* value of >0.05 by ANOVA). Finally, since previous works proposed that some *β*-lactamases may proceed from ancestors in common with penicillin binding proteins (PBPs) displaying a peptidoglycan-degrading (peptidase) capacity (28–34), it is likely that certain *β*-lactamases could retain a residual degree of this activity. In this regard, we hypothesized that our attenuated strains could display acquired weaknesses in their cell walls because of this residual activity, which would be more exaggerated in a recycling-defective background. To ascertain this possibility, we analyzed the susceptibility of our strains to (i) hypoosmotic shock (as previously described, strains with weakened peptidoglycans may display increased susceptibility to lysis associated with osmotic pressure changes [35]) and (ii) treatment with a cell wall-lytic immune element, such as lysozyme (36, 37). In fact, we described in the past that strains with a peptidoglycan recycling impairment are more susceptible to the latter attack if used together with a permeabilizing agent, such as colistin, at subinhibitory concentrations, which could also be applicable here (36). However, the results displayed in Fig. S4 indicated that none of the OXA-harboring constructs revealed a significant further increase in susceptibility to the aforementioned treatments when compared to that of strain PAΔAG. The only exception was strain PAΔAG (pUCPAC), which showed a slight but significant decrease in survival after lysozyme-plus-colistin treatment (ca. 2- to 3-fold), in accordance with previous data (36).

## DISCUSSION

Throughout this pioneering work, we assess for the first time the profiles of virulence attenuation associated with the production of an extensive set of relevant *β*-lactamases, in both *P. aeruginosa* wild-type and peptidoglycan recycling-defective backgrounds. In brief, our main findings were as follows: (i) recycling disturbance *per se* significantly decreased the *P. aeruginosa* killing capacity against *G. mellonella*; (ii) the expression of representative class A or B transferable *β*-lactamases had no impact on virulence; (iii) the expression of transferable class D OXA-2-like enzymes entailed significant levels of virulence attenuation, ranging from mild (OXA-2, only appreciable in strain PAΔAG) to extraordinarily high (with LD$_{50}$s >1,000-fold higher than that of PAO1, as happened for OXA-539); and (iv) overall, the degree of virulence loss increased in parallel with the level of peptidoglycan recycling impairment, as well as in parallel with the enzymes' hydrolytic spectra (higher attenuation associated with derived ESBLs than for OXA-2), although the molecular basis for these results remains to be elucidated.

Our results regarding recycling impairment determining *per se* a slight but significant decrease in the killing capacity of PAO1 against *G. mellonella* could be justified, at least partially, by the slightly slower exponential growth of the AmpG KO strains (Fig. S1). The likely even more impaired growth of strain PAΔAG within the animal, in which conditions are not as optimal as those in liquid culture, also supports the idea of correct cell wall recycling gaining importance to save energy and enable full virulence. Moreover, our results in the present work would also be in line with previous findings of our group in a murine model, in which mutants with AmpG or NagZ inactivation showed significantly decreased values in the virulence-related parameters studied (38).

Regarding the class A and B transferable *β*-lactamases (GES-1 and VIM-1) studied, our results are in line with previous findings in *P. aeruginosa*, in which the production of a *bla*$_{IMP}$-type class B carbapenemase barely changed the virulence of the parent strain (39), but would be in disagreement with some other studies in different species, in which the expression of other horizontal *β*-lactamases highly impaired bacterial fitness and/or pathogenesis (11). Either way, what can be deduced in short from our results is that the side effects of hyperproducing AmpC or expressing OXA-2-like enzymes, whichever they are, that end up dampening *P. aeruginosa* virulence are not a common feature for all *β*-lactamases as a whole (of course including GES-1 and VIM-1), which could explain the successful dissemination of the latter types of enzymes worldwide

(40–42). Therefore, although all $\beta$-lactamases obviously share the feature of hydrolyzing the $\beta$-lactam ring, they do not display a uniform behavior with regard to the potential attenuation of bacterial pathogenic power, even in the specific situation of peptidoglycan recycling disturbance.

As regards the OXA-2-like enzymes studied here, our results indicate that their production by *P. aeruginosa* entailed significant virulence attenuations, although with important differences depending on enzyme variant and strain background. Our data revealed a kind of positive association between the level of attenuation and the degree of peptidoglycan recycling disturbance, i.e., the greater the recycling impairment, the more reduced the killing capacity against *G. mellonella*. Thus, as a general rule, when the different OXAs were expressed in strain PAΔAG (which has a virtual recycling blockade, since AmpG is the main gate for the entrance of muropeptides to be recycled into the cytosol), the $LD_{50}$s were slightly higher than those of strain PAΔnZ (a mutant with defective recycling of sugar moieties but theoretically functional recycling of stem peptides [26]). In turn, the latter strain expressing the different OXAs was much less virulent than PAO1 under equal conditions. In the same line, the OXA-2-associated attenuating effect (which was the lowest among the tested OXAs) was only significant in the AmpG-defective background. These results were to be expected, since in a situation of energy and metabolic alteration associated with recycling impairment, impacts on features like virulence should be more exaggerated than under regular conditions (10).

Beyond the recycling background, one of the most interesting ideas that can be extracted from our results is the fact that punctual amino acid changes may affect not only the hydrolytic spectra of $\beta$-lactamases (22, 23) but, apparently, also virulence. In this regard, it is important to remember that OXA-226, OXA-161, and OXA-539 are OXA-2-derived ESBLs harboring amino acid changes very close to a proposed $\beta$-turn or loop (position 150) involved in $\beta$-lactam hydrolysis, which prompted an increased capacity for cephalosporin hydrolysis and for resistance to inhibition by tazobactam and avibactam, as demonstrated by the MICs shown in Fig. 6A (17–19). Interestingly, in this sense, the capacity for ceftazidime hydrolysis (represented by the MICs determined in strain PAO1 [Fig. 6A], which were the same in strain PAΔAG [not shown]) directly correlated to the increases in the $LD_{50}$s associated with the enzymes tested in the latter strain (Spearman's coefficient $r = 0.985$, $P = 0.0028$) (Fig. 6B). This fact suggests that the expansion in the capacity for hydrolyzing extended-spectrum cephalosporins acquired through mutations in the catalytic center of the OXA-2 enzyme (17–19) is intimately related to the derived consequences for bacterial virulence. This idea is supported by the fact that the most exaggerated change (duplication of a key catalytic residue in OXA-539 instead of a punctual amino acid substitution) entailed the highest cephalosporin hydrolysis but also the greatest decrease in the killing capacity against *G. mellonella*.

Connected to these data, the following argument could be inferred: if punctual amino acid changes notably impact the hydrolytic spectrum of the enzyme, why should they not affect other activities performed by the enzyme? In this sense, it has been proposed that certain $\beta$-lactamases display a residual peptidase activity reminiscent of their potential PBP ascendance and, therefore, could be capable of degrading the peptidoglycan or at least of affecting its physiology by cleaving certain bonds (28–34). In accordance with this, the work of Fernández et al. analyzed different acquired $\beta$-lactamases and their impacts on the fitness and peptidoglycan structure of *E. coli*, interestingly finding that two of the enzymes causing greater effects were OXA-10 and OXA-24. This could support the possibility of certain OXA-type enzymes displaying a residual peptidoglycan-degrading capacity, although it is true that no OXA-2-like enzymes were analyzed in that study (33). In any case, as previously proposed, if a $\beta$-lactamase shows a certain degree of peptidoglycan-degrading capacity (potentially increased through mutations in the catalytic center), this should have more consequences in a situation in which recycling is impaired and, thus, in which *de novo* peptidoglycan synthesis may not be enough to build a robust cell wall (36). In this scenario, the confluence of the production of the specific $\beta$-lactamase and insufficient incorporation of new material would weaken the structure to a great extent, making it more susceptible

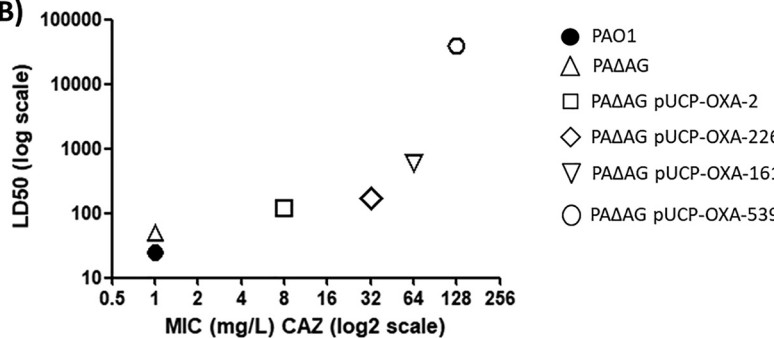

**A)**

| Strain | MIC (mg/L) | | |
|---|---|---|---|
| | TOL/TAZ | CAZ/AVI | CAZ |
| PAO1 | 0.5/4 | 0.5/4 | 1 |
| PAO1 pUCP24 | 0.5/4 | 0.5/4 | 1 |
| PAO1 pUCP-OXA-2 | 0.5/4 | 1/4 | 8 |
| PAO1 pUCP-OXA-226 | 16/4 | 16/4 | 32 |
| PAO1 pUCP-OXA-161 | 8/4 | 24/4 | 64 |
| PAO1 pUCP-OXA-539 | 16/4 | 32/4 | 128 |

**B)**

Legend:
- ● PAO1
- △ PAΔAG
- □ PAΔAG pUCP-OXA-2
- ◇ PAΔAG pUCP-OXA-226
- ▽ PAΔAG pUCP-OXA-161
- ○ PAΔAG pUCP-OXA-539

Y-axis: LD50 (log scale) — 10, 100, 1000, 10000, 100000
X-axis: MIC (mg/L) CAZ (log2 scale) — 0.5, 1, 2, 4, 8, 16, 32, 64, 128, 256

**FIG 6** (A) MICs of relevant cephalosporins/combinations in the PAO1 transformants producing OXA-2-like $\beta$-lactamases cloned in the pUCP24 vector. MICs of TOL/TAZ and CAZ/AVI were determined through commercial test strips, whereas those of CAZ alone were determined by broth microdilution. TOL/TAZ, ceftolozane/tazobactam; CAZ/AVI, ceftazidime/avibactam. (B) Graphical representation of correlation between LD$_{50}$s and ceftazidime MICs in the PAΔAG strain (as a model for peptidoglycan-recycling blockade) harboring the different OXA-2-like enzymes studied in this work. Spearman's coefficient $r$ = 0.985; $P$ = 0.0028.

to lysis and thereby dampening pathogenesis. In this sense, although we have previously shown that susceptibility to c-type lysozyme is increased in *P. aeruginosa* recycling-deficient strains (and as expected, more exaggeratedly in strain PAΔAG than in strain PAΔnZ [36, 37]), our results here suggest that the peptidoglycans of strain PAΔAG producing the different OXA enzymes are not further weakened, at least against hypoosmotic shock or c-type lysozyme attacks. In this sense, however, it is important to note that insects display a wider array of peptidoglycan-targeting proteins than mammals (genes encoding six peptidoglycan recognition proteins [PGRPs], one invertebrate-type lysozyme, and four c-type lysozymes are present in the *G. mellonella* genome [43, 44]), which means an increased variety of pathways to attack this structure by cleaving different types of bonds (45). Therefore, the possibility of these *G. mellonella*-specific cell wall-targeting weapons being responsible for the decrease in the killing capacity of the strains in which OXA residual activity would lead to weakened peptidoglycans must also be taken into account. Future work to test the susceptibility of our OXA-producing strains to specific insect cell wall-targeting humoral weapons would be needed to ascertain this option. Additionally, future experiments to translate our findings into a mammal host (a murine model of systemic infection, for instance), will help to discard these ideas and increase the potential of our study. However, the latter theoretical possibilities aside, our results suggest that the peptidoglycans of the OXA-producing strains are not clearly weakened. Thus, the above-mentioned residual activity of OXA-2-like enzymes on the entire murein sacculus would not be the most likely explanation for the virulence attenuations. However, we cannot rule out the fact that the cited residual activity might be directed not toward the intact peptidoglycan but toward its released fragments. In fact, it has been shown that certain enzymes, such as *P. aeruginosa* AmpD, AmpDh2, and AmDh3 amidases, display very different capacities for cleaving the same bonds within intact peptidoglycan versus soluble fragments (46), and therefore, it is plausible that the cited degradative activity of certain $\beta$-lactamases could affect the pool of soluble muropeptides without altering the peptidoglycan's structure and resistance. Since muropeptides have been proposed to be capable of controlling not only the expression of AmpC but also other processes, including virulence, by binding with global regulators (47–52), our results would fit perfectly with this idea. In other words, the residual activity of specific OXAs could metabolize certain soluble muropeptides with

virulence-regulating capacity within the periplasm, which would end up reaching the cytoplasm and causing the decrease in pathogenesis we report, through regulators/pathways yet to be characterized. Although a complete dissection of the molecular basis for the reported virulence attenuations was not the primary objective here, our results open the door for future lines of investigation to verify this last hypothesis. Thus, studies intended to identify/quantify the pool of soluble muropeptides (47), to describe the static structure of the sacculus (53), and to ascertain whether the expression of key virulence genes is altered (12) in our OXA-2-like-enzyme-producing strains will be needed in order to answer these questions in a very exciting and just-opened field.

Altogether, our results indicate that the production of OXA-2-derived ESBLs entails a drastic loss of virulence in *P. aeruginosa*, which may be a great handicap for the dissemination of this kind of enzymes in the clinical setting. In fact, detection of specific OXA-2-like ESBLs is only punctual for the time being in *P. aeruginosa* (17–19, 54), likely linked to resistance development during antibiotic exposure but not widely further disseminated. These circumstances contrast with those of other OXA variants with probably much reduced biological cost, such as precisely OXA-2 or OXA-10, which are among the most common transferable β-lactamases detected in *P. aeruginosa* (22, 23, 55). Related to this context, it could be of interest in the future not only to assess the levels of virulence loss associated with OXA-2-derived ESBLs but to expand the characterization to other relevant class D enzymes in *P. aeruginosa*, such as the wide array of narrow-spectrum OXAs and their ESBL derivatives, or even to some carbapenemase-hydrolyzing representatives occasionally found in clinical strains (22, 23). This would enable a better understanding of the level of conservation of our virulence attenuation model in clinical strains, revealing the therapeutic value of each enzyme and providing new insights to base future antivirulence strategies on.

Since the dissemination of OXA enzymes is an increasing concern in humans and other species (23), our results provide evidence that an imaginative strategy is to somehow learn and take advantage of the existence of these resistance determinants. At first sight, if we could design drugs causing an efficient peptidoglycan recycling blockade (in fact, some progress is currently being made, mainly with regard to NagZ inhibitors, principally intended to disable AmpC production [56, 57]), they would gain effectiveness in strains harboring OXA-2-like β-lactamases. That is to say, if these inhibitors were combined with β-lactams, this selective pressure could maintain the OXA-2-like enzyme expressed in the bacterium, collaterally reducing its virulence, in a kind of double-edged sword posing an excellent antivirulence strategy. Additionally, if we take into account the fact that peptidoglycan recycling disruption *per se* also significantly impacts *P. aeruginosa* virulence, the simple strategy of this blockade might even be enough to exert a relevant antivirulence effect against a likely interesting proportion of clinical strains even in the absence of OXA enzymes, a possibility that deserves to be explored in the future (38).

In summary, if henceforth, thanks to the fields opened though the present work, we manage to understand why certain β-lactamases are so successful at reducing the pathogenic power of *P. aeruginosa* and what the mediator pathways are, we will be one step closer to the design of more directed and effective virulence-targeting antipseudomonal strategies.

## MATERIALS AND METHODS

**Bacterial strains, plasmids, and antibiotic susceptibility testing.** A list and descriptions of the bacterial strains and plasmids used in this work are shown in Table 1. As a general rule, the cloned β-lactamases were introduced by electroporation into the following backgrounds: wild type, AmpG defective, and NagZ defective in strain PAO1 or wild type and AmpG defective in strain PA14. When indicated for selected strains, susceptibility testing to determine the MICs of representative β-lactams was performed using MIC test strips (Liofilchem) following the manufacturer's instructions and/or Müller-Hinton broth microdilution following standard procedures.

**Cloning of *bla*$_{GES-1}$ β-lactamase.** For cloning *bla*$_{GES-1}$ β-lactamase into the pUCP24 vector, the primers GES-1-F-EcoRI (TC<u>GAATTC</u>GATAAATTTCCATCTCAAGGGATC) and GES-1-R-HindIII (TC<u>AAGCTT</u>CTATTTGTCCGTG CTAGGAT) were used with the DNA of a previously described clinical *P. aeruginosa* strain (PA-A1) as the template (24) (restriction sites are underlined). The PCR products obtained were checked by sequencing, and the resulting plasmid (pUCP-GES-1) was transformed into *Escherichia coli* strain XL1-Blue through the CaCl$_2$ heat

**TABLE 1** Strains and plasmids used in this work and their relevant characteristics

| Strain or plasmid | Genotype/relevant characteristic(s)[a] | Reference or source |
|---|---|---|
| *P. aeruginosa* strains | | |
| PAO1 | Completely sequenced reference strain | 61 |
| PAΔAG | PAO1 Δ*ampG::lox*; *ampG* encodes the specific permease allowing the entry of peptidoglycan fragments into the cytosol | 21 |
| PAΔnZ | PAO1 Δ*nagZ::lox*; *nagZ* encodes the β-N-acetylglucosaminidase essential to generate 1,6-anhydromuropeptides, thought to induce AmpC production; this reaction is believed also to be essential for correct peptidoglycan recycling (specifically of sugar compounds) | 20 |
| PAjul08 | Clinical *P. aeruginosa* strain harboring $bla_{OXA-161}$ ($bla_{OXA-2}$-derived extended-spectrum class D β-lactamase) | 17 |
| PAO1(pPAjul08) | PAO1 transformed with the natural plasmid carrying the $bla_{OXA-161}$ gene | 17 |
| PA-H12O-12 | Clinical strain harboring $bla_{VIM-1}$ (class B carbapenem-hydrolyzing metallo-β-lactamase [MBL]) | 25 |
| PA-A1 | Clinical strain harboring $bla_{GES-1}$ (class A extended-spectrum β-lactamase) | 24 |
| PA14 | Completely sequenced reference strain, considered highly virulent and cytotoxic | 62 |
| PA14ΔAG | PA14 Δ*ampG::lox* | 12 |
| | | |
| Plasmids | | |
| pUCP24 | Gmʳ; pUC18-based *Escherichia-Pseudomonas* multicopy shuttle vector | 63 |
| pUCPAC | Gmʳ; pUCP24 containing PAO1 *ampC* wild-type gene | 64 |
| pUCP-GES-1 | Gmʳ; pUCP24 containing $bla_{GES-1}$ gene | This work |
| pUCP-VIM-1 | Gmʳ; pUCP24 containing $bla_{VIM-1}$ gene | 65 |
| pUCP-OXA-2 | Gmʳ; pUCP24 containing $bla_{OXA-2}$ gene (class D narrow-spectrum β-lactamase) | 17 |
| pUCP-OXA-226 | Gmʳ; pUCP24 containing $bla_{OXA-226}$ gene ($bla_{OXA-2}$-derived extended-spectrum class D β-lactamase, formerly known as OXA-144) | 18 |
| pUCP-OXA-161 | Gmʳ; pUCP24 containing $bla_{OXA-161}$ gene | 17 |
| pUCP-OXA-539 | Gmʳ; pUCP24 containing $bla_{OXA-539}$ gene ($bla_{OXA-2}$-derived extended-spectrum class D β-lactamase) | 19 |

[a]Gmʳ, gentamicin resistance cassette.

shock method. After plasmids were extracted using commercial kits, they were electroporated into the *P. aeruginosa* strains indicated, and then checked again for the absence of mutations through $bla_{GES-1}$ gene sequencing.

**Analysis of gene expression.** To verify that the cloned β-lactamases were expressed at similar levels and, therefore, that potential virulence alterations were not linked to different amounts of β-lactamase production, the mRNA of each gene was quantified through real-time RT-PCR according to previously described protocols (12, 58). Total RNA was extracted with the RNeasy minikit (Qiagen) and treated with 2 U of Turbo DNase (Ambion) for 60 min at 37°C to remove contaminating DNA. Five nanograms of purified RNA was used for one-step reverse transcription and real-time PCR using the QuantiTect SYBR green RT-PCR kit (Qiagen) in a CFX Connect device (Bio-Rad). The *rpsL* housekeeping gene was used to normalize mRNA levels using previously described primers (12, 58), and the results were referred to the results for strain PAO1 (to quantify *ampC* expression) or to the results for the respective clinical strains harboring the different β-lactamases (to quantify the expression of $bla_{VIM-1}$, $bla_{GES-1}$, or $bla_{OXA-2}$ and derivatives) (Table 1) (17, 24, 25). All RT-PCRs were performed in duplicate, and the mean expression values from three independent RNA extractions were considered. The primers used for these RT-PCRs are listed in Table 2.

**Invertebrate infection model.** The wax moth *Galleria mellonella* was used as the infection model, following previously described protocols (12, 59). Exponentially growing cultures of the corresponding strains were pelleted, washed, and resuspended in Dulbecco's phosphate-buffered saline (PBS) without calcium/magnesium (Biowest). Different serial dilutions (depending on the strain) were made in PBS and injected using Hamilton syringes (10-μl aliquots) into individual research-grade *Galleria mellonella* larvae (approximately 2-cm-long TruLarv caterpillars; Biosystems Technology) via the hindmost left proleg. Ten larvae were injected for each dilution and strain and scored as live or dead after 20 h at 37°C. An approximate 50% lethal dose ($LD_{50}$) was initially determined in a pilot screening of wide bacterial-load intervals (logarithmic scale). When an approximate $LD_{50}$ was obtained, three final experiments with already adjusted bacterial loads were performed, with the final numbers of injected bacteria verified by plating serial dilutions and colony counts. In all cases, 10 larvae were inoculated with 10 μl of PBS as controls. The percentage of larvae that died at each bacterial dose was modeled and analyzed by Probit analysis, and the $LD_{50} \pm$ standard deviation (SD) was finally determined using R software, version 3.2.2 (12). At least three independent $LD_{50}$s per strain were calculated with this model, and a final mean value $\pm$ SD was obtained and statistically analyzed as explained below.

***In vitro* growth rates.** For growth assays, standard procedures were followed (12). Briefly, 1-mL samples taken from the corresponding overnight liquid cultures were diluted in 50 mL of fresh LB broth in 250-mL flasks and incubated at 37°C and 180 rpm agitation. The doubling times of exponentially growing cells were determined by plating serial dilutions on LB agar plates at 1-h intervals. At least three independent experiments were performed for each of the selected strains.

**TABLE 2** Primers used for the analysis of gene expression in this work

| Primer | Sequence (5′–3′)[a] | PCR product size (bp) | Reference or source |
|--------|------------------|----------------------|---------------------|
| RpsL-1 | GCTGCAAAACTGCCCGCAACG | 250 | 66 |
| RpsL-2 | ACCCGAGGTGTCCAGCGAACC | | |
| AC-RNA-F | GGGCTGGCCTCGAAAGAGGAC | 246 | 67 |
| AC-RNA-R | GCACCGAGTCGGGGAACTGCA | | |
| VIM-1-RNA-F | AGATTGCCGATGGTGTTTGGT | 265 | This work |
| VIM-1-RNA-R | GATGCGTACGTTGCCACCC | | |
| GES-1-RNA-F | AGCGGTTTCTAGCATCGGGA | 260 | This work |
| GES-1-RNA-R | CATAGAGGACTTTAGCCACAG | | |
| OXA-2-RNA-F[b] | TTTTCGATGGGACGGCGTTAA | 256 | This work |
| OXA-2-RNA-R | ATAGAGCTTCCTGAGAAATGCA | | |

[a]Sequences were obtained from the published PAO1 genome or from sequences of $bla_{VIM-1}$, $bla_{GES-1}$, or $bla_{OXA-2}$ and derivatives deposited in GenBank.

[b]Since the hybridization site for these primers was conserved for OXA-2 and the three OXA-2 derivatives studied in this work, this same pair of primers was used to quantify the expression of all of them.

**Motility assays.** Swimming, swarming, and twitching motilities were determined in the selected strains as described previously (12) in plates containing different media. (i) To determine swimming motility, 10 g/liter tryptone, 5 g/liter NaCl, and 0.3% (wt/vol) mid-resolution agarose was used. Plates were inoculated with an isolated colony from an overnight culture in LB agar at 37°C, using a sterile toothpick. (ii) To determine swarming motility, 1× M8 minimal medium (60) was supplemented with 1 mM MgSO4, 0.2% glucose, 0.5% Bacto Casamino Acids, and 0.5% agar. Aliquots (2.5 $\mu$l) were taken from overnight cultures to inoculate the surface of the plate. (iii) To determine twitching motility, isolated colonies were inoculated with a sterile toothpick inserted in the bottom of the LB agar plates. In all cases, the plates were wrapped with film to prevent dehydration and incubated at 37°C for 16 h. After incubation, the diameter of the motility zone was measured. In the plates used to determine twitching motility, the medium was taken off the plate, and the print over the bottom was measured. If the motility halo was irregular, two perpendicular diameters were measured, and the result was expressed as the mean value. The PAO1 strain and derivatives were used in swimming and swarming assays, whereas the PA14 strain and derivatives were used for twitching assays, since our laboratory collection PAO1 strain is defective in this type of motility. At least 10 determinations for each strain and motility type were recorded.

**Cell culture experiments.** The A549 line of human alveolar type II pneumocytes (Sigma-Aldrich) was used between passages 3 and 30. A549 cells were maintained in RPMI 1640 medium (without phenol red) supplemented with 2 mM glutamine, 10 mM HEPES, 10% heat-inactivated fetal bovine serum, and 1× antibiotic-antimycotic solution (Biowest) at 37°C, 5% $CO_2$, and 100% relative humidity. The day before the assays, the cells were seeded at ca. $0.5 \times 10^5$ cells per well in 24-well plates. The day after, the cells were ≈90% confluent ($\approx 1 \times 10^5$ cells/well) and were infected at a multiplicity of infection (MOI) of 100, following previously described protocols (12). Briefly, bacteria from overnight liquid LB cultures were centrifuged and washed with PBS. Bacteria were then diluted in the above-described RPMI 1640 without serum or antibiotic-antimycotic solution (henceforth called infection medium [IM]): ca. $1 \times 10^7$ bacteria/500 $\mu$L of IM were used to infect each well after discarding the initial cell culture medium and washing with PBS. After an infection period of 3 h, the IM was collected and stored at −80°C for additional analyses (inflammation and cytotoxicity, see below).

To determine invasion capacity, the PAO1 strain and derivatives harboring the $\beta$-lactamases indicated (see Results) were used in classic aminoglycoside exclusion assays (12): after 3 h of infection, the IM was replaced with 1 mL/well of fresh RPMI 1640 containing 0.4 mg/mL of amikacin for 1.5 h to kill extracellular bacteria, and then this medium was removed and cells were washed twice with PBS. An amount of 0.5 mL of PBS containing 0.1% Triton X-100 (Sigma-Aldrich) was added to each well and incubated for 10 min to lyse cells and release intracellular bacteria. Serial dilutions were plated to determine the number of released CFU. Supernatants of the initial 3-h infection were used for determining the inflammatory response, using the secretion of interleukin-8 (IL-8) as an indicator, measured through a human IL-8 instant enzyme-linked immunosorbent assay (ELISA) kit (Invitrogen). To assess the cytotoxicity caused in A549 cells, the PA14 strain and derivatives harboring the $\beta$-lactamases indicated (see Results) were used, following the aforementioned infection protocol. After 3 h of infection, supernatants were processed with the cytotoxicity detection kit plus (Roche), which measures lactate dehydrogenase (LDH) release as a marker of cell death. The above-described cell culture-related determinations were always performed with samples proceeding from at least nine wells (three wells from each of three independent plates) per strain.

**Lysozyme susceptibility.** The bactericidal activity of chicken egg white lysozyme (50,000 units/mg protein, >99% protein; Sigma-Aldrich) was assessed in the indicated strains following previously described protocols (37). A total of $1 \times 10^6$ CFU of each strain, proceeding from overnight LB broth cultures, was incubated in sodium phosphate buffer (10 mM [pH 7.0]) with 25 mg/L lysozyme (in a total reaction mixture volume of 0.3 mL) for 1 h at 37°C and 180 rpm agitation and quantified by serial plating at the beginning and end of incubation (37). As previously described (37), the experiments were also performed with the addition of colistin (Sigma-Aldrich) as a permeabilizing agent, at a final subinhibitory concentration of 0.025 mg/L, in independent experiments. The effect of colistin alone on the different

strains was also studied, using the same procedure and buffer without adding lysozyme. All these experiments were performed at least in triplicate with the indicated strains.

**Osmotic shock susceptibility.** The resistance of the indicated strains against hypoosmotic shock was assessed following published protocols (37). Overnight LB broth cultures were centrifuged, washed three times with double-distilled water, and afterwards resuspended in a final volume of 10 mL of double-distilled water at a final concentration of $1 \times 10^5$ CFU/mL. The suspensions were incubated at room temperature for 24 h with gentle agitation. Bacterial viability was quantified through colony counts after serial dilutions and plating at the beginning and the end of incubation. The assay was performed in independent triplicates per strain.

**Statistical analysis.** With the exception of $LD_{50}$s (see above), GraphPad Prism 5 software was used for statistical analysis and graphical representation. Quantitative variables were analyzed through repeated-measures ANOVA (with the *post hoc* Tukey's multiple-comparison test), pairing data obtained from the experimental replicates (i.e., matched observations), and/or Student's *t* test (two-tailed, paired) as appropriate. The statistical correlations indicated were checked through Pearson's or Spearman's rank coefficients as appropriate. A *P* value of <0.05 was considered statistically significant.

**Data availability.** The data sets generated for this study are available upon request from the corresponding authors.

## SUPPLEMENTAL MATERIAL

Supplemental material is available online only.

**SUPPLEMENTAL FILE 1**, PDF file, 0.8 MB.

## ACKNOWLEDGMENTS

This work was financed by the Balearic Islands Government grants FPI/2206/2019 and FOLIUM17/04, the Spanish Network for Research in Infectious Diseases (REIPI, RD16/0016/0004), and grants number CP12/03324, PI15/00088, PI15/02212, CPII17/00017, PI18/00076, PI18/00681, PI21/00753, and FI19/00004 from the Instituto de Salud Carlos III (Ministerio de Ciencia e Innovación, Spain) and cofinanced by the European Regional Development Fund A way to achieve Europe.

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
