## [Reviewer comments · Microbiology Spectrum]

Microbiology Spectrum

Impact of peptidoglycan recycling blockade and expression of horizontally-acquired β -lactamases on *Pseudomonas aeruginosa* virulence.

Isabel Maria Barceló, Gabriel Torrens, Maria Escobar-Salom, Elena Jordana, María Magdalena Capó Bauzá, Carlos Ramón Pallín, Daniel García, Pablo Fraile-Ribot, Xavier Mulet, Antonio Oliver, and Carlos Juan

Corresponding Author(s): Carlos Juan, Institut d'Investigació Sanitaria de Balears

Review Timeline:

Submission Date:	November 12, 2021
Editorial Decision:	December 21, 2021
Revision Received:	December 23, 2021
Accepted:	January 24, 2022

Editor: Pablo Power

Reviewer(s): Disclosure of reviewer identity is with reference to reviewer comments included in decision letter(s). The following individuals involved in review of your submission have agreed to reveal their identity: Jose Manuel Rodriguez-Martínez (Reviewer #2)

Transaction Report:

DOI: <https://doi.org/10.1128/spectrum.02019-21>

December 21, 2021

Dr. Carlos Juan
Institut d'Investigació Sanitària de Balears
Palma de Mallorca
Spain

Re: Spectrum02019-21 (Impact of peptidoglycan recycling blockade and expression of horizontally-acquired β -lactamases on *Pseudomonas aeruginosa* virulence.)

Dear Dr. Carlos Juan:

Link Not Available

Sincerely,

Pablo Power

Journals Department
Reviewer comments:

Reviewer #1 (Comments for the Author):

1- In this study, the authors showed that the AmpC overexpression without peptidoglycan recycling blockade did not affect the virulence of *P. aeruginosa*. Can the authors provide a discussion regarding this issue.

2- It would be interesting to know the LD50 of PA Δ nZ harboring GES and VIM.

3- In the figure S3, the authors have analyzed the susceptibility of PAO1 and PA Δ AG harboring OXA-2, OXA-226, OXA-161 and OXA-539 to osmotic shock and lysozyme and concluded that none of the OXA-harboring constructs revealed a significant increase in the susceptibility to osmotic shock and lysozyme. When I see the results of the figure 3, the higher LD50 have been obtained with the mutant PA Δ nZ harboring different OXA. It would be interesting to provide data on the susceptibility of PAO1 and PA Δ nZ harboring OXA-2, OXA-226, OXA-161 and OXA-539 to osmotic shock and lysozyme.

4- Lines 377-392: This paragraph is not completely true because some control experiments are missing. I think that results of the susceptibility of PAO1 and PA Δ nZ harboring OXA-2, OXA-226, OXA-161 and OXA-539 to osmotic shock and lysozyme

should confirmed or not the hypothesis of the authors.

Reviewer #2 (Comments for the Author):

Dear Editor,

Pseudomonas aeruginosa is one of the nosocomial pathogens whose requires therapeutic options extremely urgent. In this study, an interesting point of view related to the resistance-virulence interplay is analyzed as source of therapeutic strategies. In this regard, authors observe virulence attenuations associated with chromosomal (*ampC*) and transferable β -lactamases like OXA-2-derived extended-spectrum β -lactamases and its correlation with mutations causing the expansion of their hydrolytic spectrums. That was also associated to peptidoglycan recycling disturbance as a weapon to attenuate *Pseudomonas aeruginosa* virulence in class C and D β -lactamase production backgrounds. Unfortunately, in this work it was not possible to find a mechanistic explanation for these interesting results, although different possibilities are discussed. In my opinion, this work deserve attention and collects information of interest to develop future virulence-targeting antipseudomonal strategies.

On the other hand, I consider that the current manuscript format could be partially shortened, overall, the Discussion section. Sometimes this section is repetitive and its extension go beyond than necessary. That said, the work is very well written in an easy-to-read way and with understandable language in most of the manuscript. The methods used are appropriate in relation to the results and conclusions drawn.

In my opinion, the manuscript contains some deficiencies or issues that can be improved during this revision process:

1) Although the authors have quantified the expression of the genes evaluated by qPCR (and it is evident that the expression of these genes in the cloning plasmids is higher than that obtained from their natural plasmids), I think it is important to reflect in the manuscript that the results obtained in relation to the attenuation of virulence (higher LD50) could in part be artifact for this reason. One reason for this is the multicopy character of the selected vector (pUCP24), in relation to the number of copies expected in a natural plasmid that encodes this type of genes.

2) Additionally, in relation to comment 1, another aspect of interest would be to test how natural plasmids affect virulence. In this case, for example, I would have liked to see how a natural OXA-539-producing plasmid affects virulence in a deficient *ampG* background.

2) Another aspect to mention is the limited methodology to assess virulence. In this case, a single reference value such as LD50 was used in a *Galleria mellonella* model. Without a doubt, it would have been interesting to evaluate in this work (at least for the most interesting combinations) the reproducibility of these results in a mouse animal mode (of peritoneal sepsis as a suggestion). Confirmation of these results in a mammalian model would notably increase the interest of these results.

3) Under the control of which promoter are these genes in the vector pUCP24? Is it an inducible promoter? I don't think so, but if it were the case, another question of interest, beyond the type of beta-lactamase or specific mutations in them, would be interesting to determine if there is a relationship between the level of expression of these beta-lactamases (*ampC*, OXA-2 and derived) with the degree of virulence. In this case, dose / response experiments would be greatly appreciated.

4) It must be taken into account that the measurement of virulence attenuation derived from the expression of these beta-lactamases genes in *P. aeruginosa* does not allow to discern the exact impact of these genes since in all cases the *Pseudomonas* variants used have a copy of the chromosomal *AmpC* gene. Thus, it would have been more elegant to evaluate in parallel the impact on virulence in an isogenic strain of PAO1 deficient in *ampC*. I am not requesting that these experiments be performed but yes that this aspect of the work be discussed.

5) I find it strange that the LD50 values indicated in the text are always exact numbers (L128 and L201-207 versus 225, 173, 4300, 25000, ...). If I have understood correctly, these numbers are the average of several assays and could perhaps be expressed with the corresponding decimals. Maybe there is something I don't understand.

6) Several of the results shown did not provide differences between the evaluated mutants (FIG5, FIG6B), this does not mean that they are not interesting or necessary, but when reducing the length of the manuscript they could be turned to supplementary material.

Staff Comments:

Preparing Revision Guidelines

Please return the manuscript within 60 days; if you cannot complete the modification within this time period, please contact me. If you do not wish to modify the manuscript and prefer to submit it to another journal, please notify me of your decision immediately so that the manuscript may be formally withdrawn from consideration by Microbiology Spectrum.

Reviewer comments:

First, we want to thank Reviewers for the time taken to revise our manuscript, and for the interesting comments and observations, that we will try to answer/satisfy below each question (in red color).

Reviewer #1 (Comments for the Author):

1- In this study, the authors showed that the AmpC overexpression without peptidoglycan recycling blockade did not affect the virulence of *P. aeruginosa*. Can the authors provide a discussion regarding this issue.

We already showed and discussed in a previous paper (Pérez-Gallego M. et al, mBio 2016, 7: e01783-16), that the overexpression of this intrinsic β -lactamase (at ca. 1000-fold compared to wildtype, proceeding from the multicopy plasmid pUCPAC), did not have any impact *per se* over *Pseudomonas aeruginosa* (PA) fitness / virulence. In our opinion, the fact that the over-expression of AmpC in clinical strains is the main mechanism of resistance of this species (regardless of the different mutational pathways involved), suggests that the biological cost for the bacterium must be minor. In other words, if the AmpC hyper-production entailed a significant handicap for PA fitness and virulence, it would not be such an extraordinarily frequent resistance mechanism. In fact, the double mutational inactivation of *ampD* and *dacB*, eventually driving to even higher levels of AmpC hyper-production (ca. 2000-fold compared to wildtype) in several clinical strains has been reported (Zamorano L. et al, Antimicrob Agents Chemother 2011, 55: 1990–1996; Del Barrio-Tofiño E. et al, Antimicrob Agents Chemother 2017, 61:e01589-17), which supports once more the idea of a minor biological cost associated to these high levels of expression of this resistance mechanism. Moreover, we propose that the explanation for this despicable biological cost apparently associated to AmpC hyper-production *per se* may reside in the fact that, whichever is the collateral impact that the overproduction of this enzyme has in PA virulence, is only visible when its peptidoglycan (PGN) metabolism is already altered (through AmpG or NagZ disruption, for instance). In other words, if there is not a basal level of PGN recycling disturbance, the effects of AmpC are not appreciable. In fact, these circumstances are mentioned in the introduction of the first version of the paper (lines 81-84).

2- It would be interesting to know the LD50 of PAnZ harboring GES and VIM. Our data suggest that the background in which the biological burden of expressing any β -lactamase is more exaggerated, is that defective in AmpG. This is for instance easily seen in the case of pUCPOXA-2, that did not entail a significant increase in the LD50 in PAO1 or PAnZ strains, but only in PAAG. Similarly, the increases of LD50s were always higher in PAAG strain (compared to those obtained in PAnZ), when expressing whichever OXA-2-derived ESBL. This can be seen in the Fig 3, with the following approximate values of LD50s:

PAAG pUCPOXA-2: \approx 123 Colony forming units (CFUs); PAnZ pUCPOXA-2: \approx 43;

PAAG pUCPOXA-226: \approx 173 CFUs; PAnZ pUCPOXA-226: \approx 102;

PAAG pUCPOXA-161: \approx 590 CFUs; PAnZ pUCPOXA-161: \approx 500;

PAAG pUCPOXA-539: \approx 38500 CFUs ;PAnZ pUCPOXA-539: \approx 25000;

These results were fairly expectable, since the degree of PGN recycling disturbance is theoretically greater after disruption of AmpG (the main gate for the entrance of PGN fragments into cytosol), that therefore virtually blocks all the recycling pathways, than after deletion of NagZ (which is theoretically only involved in the recycling of sugar moieties, but not the stem peptides) (Johnson JW, et al, 2013; Annals of the New York Academy of Sciences, 1277:54–75).

In the light of these data and previous knowledge, we chose PAAG as the model background in which assess the biological cost of expressing a given β -lactamase. In other words, if in this background the β -lactamase does not show an associated increase in the LD50, it is highly likely that it will not show a cost in the wildtype or NagZ-deficient strains. Obviously, if a LD50 increase had been seen in PAAG pUCPVIM or PAAG pUCPGES, we would have tested the PAnZ and wildtype backgrounds, but leaning on which what we explained (and on the absence of attenuation in PAAG pUCPVIM-1/pUCPGES-1), we believe that it is not necessary. Moreover, as explained in the first answer for the case of AmpC hyper-production, since these enzymes (VIM-1 and GES-1) are worldwide disseminated in PA, it is also expectable that their expression entails a minimum biological cost, which supports our focusing.

3- In the figure S3, the authors have analyzed the susceptibility of PAO1 and PAAG harboring OXA-2, OXA-226, OXA-161 and OXA-539 to osmotic shock and lysozyme and concluded that none of the OXA-harboring constructs revealed a significant increase in the susceptibility to osmotic shock and lysozyme. When I see the results of the figure 3, the higher LD50 have been obtained with the mutant

PA Δ nZ harboring different OXA. It would be interesting to provide data on the susceptibility of PAO1 and PA Δ nZ harboring OXA-2, OXA-226, OXA-161 and OXA-539 to osmotic shock and lysozyme.

First, we want to clarify that the reviewer probably misinterpreted the data in Figure 3, since he states that PAnZ background showed the highest LD50s when expressing the different OXAs. But actually, the strain with more attenuated virulence was PAAG (please see LD50 values in the answer of question 2). Thus, for the same reasons that we have just explained in question 2, since PAAG was the background in which the virulence impairments were more exaggerated, we assume that if in this strain there were not increases in the susceptibility to lytic aggressions over the PGN, the other backgrounds would not show this kind of increased sensitivities, either. For this reason we considered that was enough to determine the susceptibility to lysozyme/osmotic shock in the PAAG background.

Moreover, in our previous paper (Torrens G. et al, PLoS One. 2017 12:e0181932), we already showed that PAAG was more susceptible to lysozyme (but also to other lytic aggressions such as that performed by Peptidoglycan Recognition Protein 2) than PAnZ, and thus, we honestly believe that determining osmotic shock / lysozyme susceptibility in the rest of backgrounds would not provide any more interesting information.

4- Lines 377-392: This paragraph is not completely true because some control experiments are missing. I think that results of the susceptibility of PAO1 and PA Δ nZ harboring OXA-2, OXA-226, OXA-161 and OXA-539 to osmotic shock and lysozyme should confirmed or not the hypothesis of the authors.

This issue is directly related with the just given answer for the question 3; therefore we believe that determining these parameters in wildtype or PAnZ backgrounds would not give any interesting data. In any case, as can be checked in the new version, we have slightly changed the writing of original lines 377-392, focusing them on PAAG background (which is the one chosen and tested), and emphasizing the fact that PAnZ has been previously shown to be more resistant than PAAG to the mentioned lytic aggressions.

Reviewer #2 (Comments for the Author):

Dear Editor,

Pseudomonas aeruginosa is one of the nosocomial pathogens whose requires therapeutic options extremely urgent. In this study, an interesting point of view related to the resistance-virulence interplay is analyzed as source of therapeutic strategies. In this regard, authors observe virulence attenuations associated with chromosomal (*ampC*) and transferable β -lactamases like OXA-2-derived extended-spectrum β -lactamases and its correlation with mutations causing the expansion of their hydrolytic spectrums. That was also associated to peptidoglycan recycling disturbance as a weapon to attenuate *Pseudomonas aeruginosa* virulence in class C and D β -lactamase production backgrounds. Unfortunately, in this work it was not possible to find a mechanistic explanation for these interesting results, although different possibilities are discussed. In my opinion, this work deserve attention and collects information of interest to develop future virulence-targeting antipseudomonal strategies.

On the other hand, I consider that the current manuscript format could be partially shortened, overall, the Discussion section. Sometimes this section is repetitive and its extension go beyond than necessary. That said, the work is very well written in an easy-to-read way and with understandable language in most of the manuscript. The methods used are appropriate in relation to the results and conclusions drawn.

Following the comment of the reviewer 2, we have shortened some paragraphs of the discussion to avoid repetitions, as can be checked in the new version.

In my opinion, the manuscript contains some deficiencies or issues that can be improved during this revision process:

1) Although the authors have quantified the expression of the genes evaluated by qPCR (and it is evident that the expression of these genes in the cloning plasmids is higher than that obtained from their natural plasmids), I think it is important to reflect in the manuscript that the results obtained in relation to the attenuation of virulence (higher LD50) could in part be artifact for this reason. One reason for this is the multicopy character of the selected vector (pUCP24), in relation to the number of copies expected in a natural plasmid that encodes this type of genes.

As can be checked in multiple previous works, the levels of AmpC hyperproduction achieved through pUCPAC are similar to those associated to mutational events (for

instance through the abovementioned double inactivation of *ampD* and *dacB*) also found in clinical strains (Moya B et al, PLoS Pathog. 2009, 5:e1000353; Zamorano L. et al, Antimicrob Agents Chemother 2011, 55: 1990–1996; Del Barrio-Tofiño E. et al, Antimicrob Agents Chemother 2017, 61:e01589-17). Therefore, we believe that the levels of *ampC* hyperexpression we use in our work are not aberrant at all, and represent quite accurately what happens in the clinical context, and thus, the AmpC overproduction-associated biological burden (seen only in PAAG or PAnZ backgrounds) is not likely an artifact.

A similar assumption could be done regarding the expression of pUCP with the different horizontally acquired Beta-lactamases cloned: as shown in the Fig 2B and 4, the levels of expression of each enzyme compared to clinical strains harboring the same resistance genes were not very different; To illustrate this, we can have into account that for instance, AmpC can be hyper-produced hundreds or even thousands-fold (through mutational mechanisms) compared to basal expression in clinical strains, which is translated into dramatic changes of MICs for Beta-lactams. Meanwhile, in the case of our GES, VIM and OXA enzymes, the differences between the expression of our constructs and the clinical strains are always below 4-fold (Fig 2B and 4), suggesting that this range of differences should not have a dramatic impact over virulence. In the same sense, in the specific case of OXA-161, we also determined the level of expression of PAO1 with the natural plasmid containing this enzyme (pPAjul08) (Juan C et al, Antimicrob Agents Chemother. 2009;53:5288-90), and was virtually the same (less than 2-fold of difference) compared to that proceeding from pUCP-OXA-161 or that from the natural clinical strain, which supports our argumentation. Therefore, we believe that our results are not an artifact, since we achieved (through pUCP24) levels of expression of AmpC, GES, VIM and OXA genes similar to those found in natural strains.

2) Additionally, in relation to comment 1, another aspect of interest would be to test how natural plasmids affect virulence. In this case, for example, I would have liked to see how a natural OXA-539-producing plasmid affects virulence in a deficient *ampG* background.

Related with what we just answer in question 1, we determined the LD50 of PAO1 strain harboring the natural plasmid in which OXA-161 is encoded (pPAjul08) (Juan C et al, Antimicrob Agents Chemother. 2009;53:5288-90). As can be seen in the Fig 3C, the strain PAO1 pPAjul08 displayed a similar increase in the LD50 than the strain PAO1 pUCP-OXA-161, which demonstrates the biological burden associated to the expression of this enzyme, even in a wildtype background. If this burden is shown in a strain with functional PGN recycling, it is obvious that this should be more exaggerated in an AmpG or NagZ-deficient strain, as we show in the same Fig with

the pUCP-OXA-161 plasmid. Therefore, we believe that this question is already answered in our manuscript.

Regarding OXA-539 specifically we could not perform the interesting suggested experiment with an alleged natural plasmid containing the *bla*OXA-539 gene, since apparently, this resistance determinant was integrated into the bacterial chromosome (Fraile-Ribot et al; Antimicrob Agents Chemother. 2017;61(9):e01117-17).

2) Another aspect to mention is the limited methodology to assess virulence. In this case, a single reference value such as LD50 was used in a *Galleria mellonella* model. Without a doubt, it would have been interesting to evaluate in this work (at least for the most interesting combinations) the reproducibility of these results in a mouse animal model (of peritoneal sepsis as a suggestion). Confirmation of these results in a mammalian model would notably increase the interest of these results.

We have added a sentence in the discussion (lines 392-4 of the Marked Up document) to emphasize that our results should be translated in future into a murine model for instance, to be fully validated and therefore gain therapeutic implications and potential.

3) Under the control of which promoter are these genes in the vector pUCP24? Is it an inducible promoter? I don't think so, but if it were the case, another question of interest, beyond the type of beta-lactamase or specific mutations in them, would be interesting to determine if there is a relationship between the level of expression of these beta-lactamases (*ampC*, OXA-2 and derived) with the degree of virulence. In this case, dose / response experiments would be greatly appreciated.

Promoter of pUCP24 vector is not inducible, and therefore, although the observation is very interesting, the proposed assays cannot be done; In any case, as abovementioned, we believe that the achieved levels of expression both for AmpC but also for the horizontal enzymes are quite representative of what happens in clinical strains, thus validating the results for the associated virulence attenuations.

4) It must be taken into account that the measurement of virulence attenuation derived from the expression of these beta-lactamases genes in *P. aeruginosa* does not allow to discern the exact impact of these genes since in all cases the *Pseudomonas* variants used have a copy of the chromosomal AmpC gene. Thus, it would have been more elegant to evaluate in parallel the impact on virulence in an

isogenic strain of PAO1 deficient in ampC. I am not requesting that these experiments be performed but yes that this aspect of the work be discussed.

While the observation is very interesting, and the use of an AmpC-deficient mutant could provide a “cleaner” image of the specific burden associated to the other enzymes, we also believe that is more realistic to use a strain that has a functional *ampC*. This is because in clinical strains, the presence of this enzyme is universal, and therefore, those strains that harbor horizontally acquired enzymes, always express them in addition to AmpC. Moreover, since we used PAO1 strain (that harbors no mutations driving to hyper expression), the constitutive level of AmpC production is very low and therefore, its impact over the final results of virulence attenuation is likely minor.

5) I find it strange that the LD50 values indicated in the text are always exact numbers (L128 and L201-207 versus 225, 173, 4300, 25000, ...). If I have understood correctly, these numbers are the average of several assays and could perhaps be expressed with the corresponding decimals. Maybe there is something I don't understand.

The reviewer is right in the sense that the values we display in the text are not the exact results of LD50s; Values are calculated through a Probit model in three independent replicates, and after that, a mean and SD are obtained, which are obviously almost never exact numbers. However, as can be checked in the manuscript, we introduced the concept “ca.” to express that these are only approximate numbers. In fact, we believe that the values we provide are perhaps even too specific, since the Figures could be explanatory by themselves. We only introduced the approximate values in the text to make easier the interpretation of Figures, and make easier for the reader to appreciate the differences between LD50s, that sometimes, with log scale are not too clear. On the other hand, we honestly believe that introducing decimals within the text for such high values of a parameter as is LD50, is unnecessary and could make the reading a bit tedious. In any case, as stated in the manuscript (“Data availability”), the datasets containing the exact values generated during the study, are available upon request.

6) Several of the results shown did not provide differences between the evaluated mutants (FIG5, FIG6B), this does not mean that they are not interesting or necessary, but when reducing the length of the manuscript they could be turned to supplementary material.

As suggested, we incorporated Fig 5 into supplementary material, although in the case of Fig 6B, since its results are intimately linked to Fig 6A (PA14 strain), we think that it is better to maintain the figure in its original form.

January 24, 2022

Dr. Carlos Juan
Institut d'Investigació Sanitaria de Balears
Palma de Mallorca
Spain

Re: Spectrum02019-21R1 (Impact of peptidoglycan recycling blockade and expression of horizontally-acquired β -lactamases on *Pseudomonas aeruginosa* virulence.)

Dear Dr. Carlos Juan:

Your manuscript has been accepted, and I am forwarding it to the ASM Journals Department for publication. You will be notified when your proofs are ready to be viewed.

Sincerely,

Pablo Power
Editor, Microbiology Spectrum
